# PRIORITIZE ALIGNMENT IN DATASET DISTILLATION

## ABSTRACT

Dataset Distillation aims to compress a large dataset into a significantly more compact, synthetic one without compromising the performance of the trained models. To achieve this, existing methods use the agent model to extract information from the target dataset and embed it into the distilled dataset. Consequently, the quality of extracted and embedded information determines the quality of the distilled dataset. In this work, we find that existing methods introduce misaligned information in both information extraction and embedding stages. To alleviate this, we propose Prioritize Alignment in Dataset Distillation (**PAD**), which aligns information from the following two perspectives. 1) We prune the target dataset according to the compressing ratio to filter the information that can be extracted by the agent model. 2) We use only deep layers of the agent model to perform the distillation to avoid excessively introducing low-level information. This simple strategy effectively filters out misaligned information and brings non-trivial improvement for mainstream matching-based distillation algorithms. Furthermore, built on trajectory matching, **PAD** achieves remarkable improvements on various benchmarks, achieving state-of-the-art performance.

## 1 INTRODUCTION

Dataset Distillation (DD) (Wang et al., 2020) aims to compress a large dataset into a small synthetic dataset that preserves important features for models to achieve comparable performances. Ever since being introduced, DD has gained a lot of attention because of its wide applications in practical fields such as privacy preservation (Dong et al., 2022; Yu et al., 2023), continual learning (Masarczyk & Tautkute, 2020; Rosasco et al., 2021), and neural architecture search (Jin et al., 2018; Pasunuru & Bansal, 2019).

Recently, matching-based methods (Zhao & Bilen, 2021c; Wang et al., 2022; Du et al., 2022) have achieved promising performance in distilling high-quality synthetic datasets. Generally, the process of these methods can be summarized into two steps: (1) *Information Extraction*: an agent model is used to extract important information from the target dataset by recording various metrics such as gradients (Zhao et al., 2020), distributions (Zhao & Bilen, 2021a), and training trajectories (Cazenavette et al., 2022), (2) *Information Embedding*: the synthetic samples are optimized to incorporate the extracted information, which is achieved by minimizing the differences between the same metric calculated on the synthetic data and the one recorded in the previous step.

In this work, we first reveal both steps will introduce misaligned information, which is redundant and potentially detrimental to the quality of the synthetic data. Then, by analyzing the cause of this misalignment, we propose alleviating this problem through the following two perspectives.

Typically, in the *Information Extraction* step, most distillation methods allow the agent model to see all samples in the target dataset. This means information extracted by the agent model comes from samples with various difficulties (see Figure 1a). However, according to previous study (Guo et al., 2023), information related to easy samples is only needed when the compression ratio is high. This misalignment leads to the sub-optimal of the distillation performance.

To alleviate the above issue, we first use data selection methods to measure the difficulty of each sample in the target dataset. Then, during the distillation, a data scheduler is employed to ensure only data whose difficulty is aligned with the compression ratio is available for the agent model.

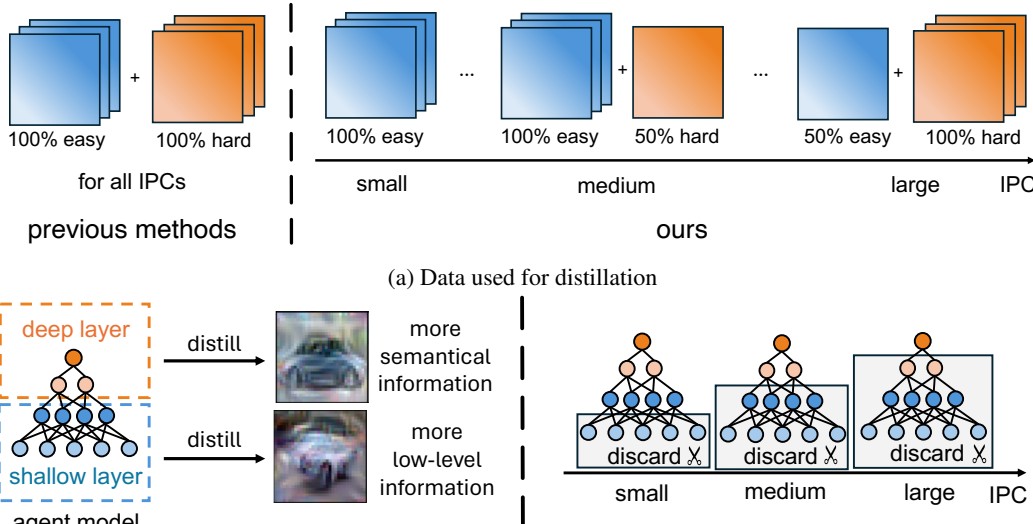

(a) Data used for distillation

(b) Parameters used for distillation

Figure 1: (a) Compared with using all samples without differentiation in IPCs (left), PAD meticulously selects a subset of samples for different IPCs to align the expected difficulty of information required (right). (b) Different layers distill different patterns (left). PAD masks out (grey box) shallow-layer parameters during metric matching in accordance with IPCs (right).

In the *Information Embedding* step, most distillation methods except DM (Zhao & Bilen, 2021a) choose to use all parameters of the agent model to perform the distillation. Intuitively, this will ensure the information extracted by the agent model is fully utilized. However, we find shallow layer parameters of the model can only provide low-quality, basic signals, which are redundant for dataset distillation in most cases. Conversely, performing the distillation with only parameters from deep layers will yield high-quality synthetic samples. We attribute this contradiction to the fact that deeper layers in DNNs tend to learn higher-level representations of input data (Mahendran & Vedaldi, 2016; Selvaraju et al., 2016).

Based on our findings, to avoid embedding misaligned information in the *Information Embedding* step, we propose to use only parameters from deeper layers of the agent model to perform distillation, as illustrated in Figure 1b. This simple change brings significant performance improvement, showing its effectiveness in aligning information.

Through experiments, we validate that our two-step alignment strategy is effective for distillation methods based on matching gradients (Zhao et al., 2020), distributions (Zhao & Bilen, 2021a), and trajectories (Cazenavette et al., 2022). Moreover, by applying our alignment strategy on trajectory matching (Cazenavette et al., 2022; Guo et al., 2023), we propose our novel method named Prioritize Alignment in Dataset Distillation (PAD). After conducting comprehensive evaluation experiments, we show PAD achieves state-of-the-art (SOTA) performance.

## 2 MISALIGNED INFORMATION IN DATASET DISTILLATION

Generally, we can summarize the distillation process of matching-based methods into the following two steps: (1) *Information Extraction*: use an agent model to extract essential information from the target dataset, realized by recording metrics such as gradients (Zhao et al., 2020), distributions (Zhao & Bilen, 2021a), and training trajectories (Cazenavette et al., 2022), (2) *Information Embedding*: the synthetic samples are optimized to incorporate the extracted information, realized by minimizing the differences between the same metric calculated on the synthetic data and the one recorded in the first step.

In this section, through analyses and experimental verification, we show the above two steps both will introduce misaligned information to the synthetic data.

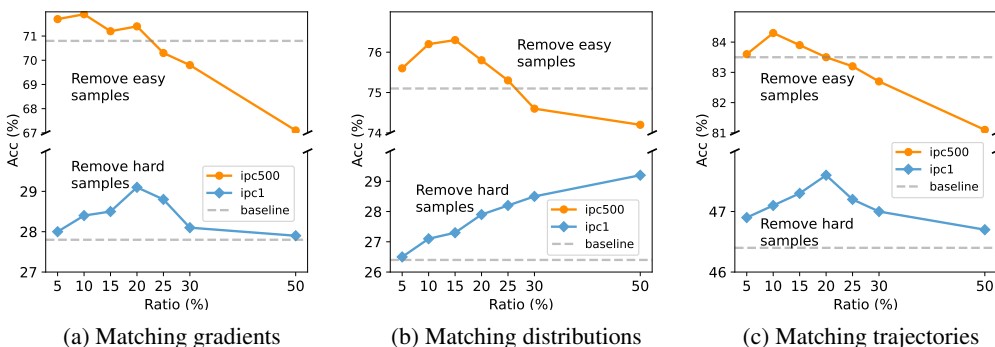

Figure 2: Distillation performance on CIFAR-10 where data points are removed with different ratios. Removing unnecessary data points helps to improve the performance of methods based on matching gradients, distributions, and trajectories, both in low and high IPC cases.

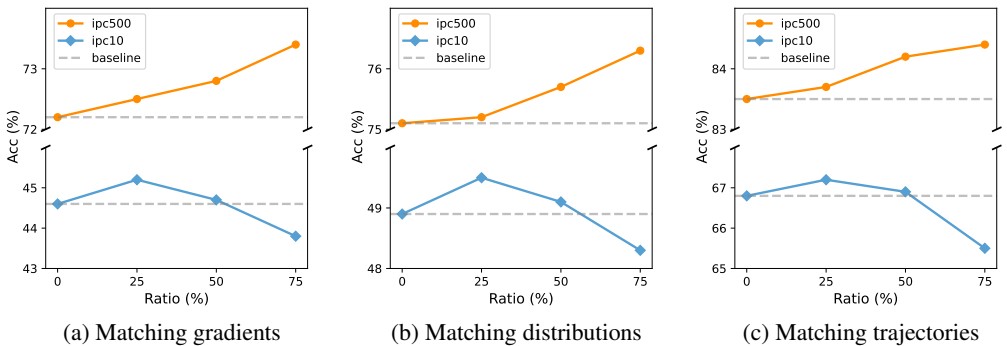

Figure 3: Distillation performances on CIFAR-10 where n% (ratio) shallow layer parameters are not utilized during distillation. Discarding shallow-layer parameters is beneficial for methods based on matching gradients, distributions, and trajectories, both in low and high IPC cases.

### 2.1 MISALIGNED INFORMATION EXTRACTED BY AGENT MODELS

In the *information extraction* step, an agent model is employed to extract information from the target dataset. Generally, most existing methods (Cazenavette et al., 2022; Du et al., 2022; Zhao et al., 2020; Zhao & Bilen, 2021c) allow the agent model to see the full dataset. This implies that the information extracted by the agent model originates from samples with diverse levels of difficulty. However, the expected difficulty of distilled information varies with changes in IPC: smaller IPCs prefer easier information while larger IPCs should distill harder one (Guo et al., 2023).

To verify if this misalignment will influence the quality of synthetic data, we perform the distillation where hard/easy samples of target dataset are removed with various ratios. As the results reported in Figure 2, pruning unaligned data points is beneficial for all matching-based methods. This proves the misalignment indeed will influence the distillation performance and can be alleviated by filtering out misaligned data from the target dataset.

### 2.2 MISALIGNED INFORMATION EMBEDDED BY METRIC MATCHING

Most existing methods use all parameters of the agent model to compute the metric used for matching. Intuitively, this helps to improve the distillation performance, since in this way all information extracted by the agent model will be embedded into the synthetic dataset. However, since shallow layers in DNNs tend to learn basic distributions of data (Mahendran & Vedaldi, 2016; Selvaraju et al., 2016), using parameters from these layers can only provide low-level signals that turned out to be redundant in most cases.

As can be observed in Figure 3, it is evident that across all matching-based methods, the removal of shallow layer parameters consistently enhances performance, regardless of the IPC setting. This proves employing over-shallow layer parameters to perform the distillation will introduce misaligned information to the synthetic data, compromising the quality of distilled data.

## 3 METHOD

To alleviate the information misalignment issue, based on trajectory matching (TM) (Cazenavette et al., 2022; Guo et al., 2023), we propose Prioritizing Alignment in Dataset Distillation (PAD). PAD can also be applied to methods based on matching gradients (Zhao et al., 2020) and distributions (Zhao & Bilen, 2021a), which are introduced in Appendix A.1.

### 3.1 PRELIMINARY OF TRAJECTORY MATCHING

Following the two-step procedure, to extract information, TM-based methods (Cazenavette et al., 2022; Guo et al., 2023) first train agent models on the real dataset $\mathcal{D}_R$ and record the changes of the parameters. Specifically, let $\{\theta_t^*\}_0^N$ be an expert trajectory, which is a parameter sequence recorded during the training of agent model. At each iteration of trajectory matching, $\theta_t^*$ and $\theta_{t+M}^*$ are randomly selected from expert trajectories as the start and target parameters.

To embed the information into the synthetic data, TM methods minimize the distance between the expert trajectory and the student trajectory. Let $\hat{\theta}_t$ denote the parameters of the student agent model trained on synthetic dataset $\mathcal{D}_S$ at timestep $t$. The student trajectory progresses by doing gradient descent on the cross-entropy loss $l$ for $N$ steps:

$$\hat{\theta}_{t+i+1} = \hat{\theta}_{t+i} - \alpha \nabla l(\hat{\theta}_{t+i}, \mathcal{D}_S), \tag{1}$$

Finally, the synthetic data is optimized by minimizing the distance metric, which is formulated as:

$$\mathcal{L} = \frac{||\hat{\theta}_{t+N} - \theta_{t+M}^*||}{||\theta_{t+M}^* - \theta_t^*||}. \tag{2}$$

### 3.2 FILTERING INFORMATION EXTRACTION

In section 2.1, we show using data selection to filter out unmatched samples could alleviate the misalignment caused in *Information Extraction* step. According to previous work (Guo et al., 2023), TM-based methods prefer easy information and choose to match only early trajectories when IPC is small. Conversely, hard information is preferred by high IPCs and they match only late trajectories. Hence, we should use easy samples to train early trajectories, while late trajectories should be trained with hard samples. To realize this efficiently, we first use the data selection method to measure the difficulty of samples contained in the target dataset. Then, during training expert trajectories, a scheduler is implemented to gradually incorporate hard samples into the training set while excluding easier ones.

**Difficulty Scoring Function** Identifying the difficulty of data for DNNs to learn has been well studied in data selection area (Mirzasoleiman et al., 2019; Killamsetty et al., 2020; 2021; Sorscher et al., 2022). For simplicity consideration, we use Error L2-Norm (EL2N) score Paul et al. (2021) as the metric to evaluate the difficulty of training examples (other metrics can also be chosen, see Section 4.3.2). Specifically, let $x$ and $y$ denote a data point and its label, respectively. Then, the EL2N score can be calculated by:

$$\chi_t(x, y) = \mathbb{E}||p(w_t, x) - y||_2. \tag{3}$$

where $p(w_t, x) = \sigma(f(w_t, x))$ is the output of a model $f$ at training step $t$ transformed into a probability distribution. In consistent with Sorscher et al. (2022), samples with higher EL2N scores are considered as harder samples in this paper.

**Scheduler** The scheduler can be divided into the following stages. Firstly, the hardest samples are removed from the training set, ensuring that it exclusively comprises data meeting a predetermined

| Dataset | CIFAR-10 | | | | | CIFAR-100 | | | | TinyImageNet | | |
|---|---|---|---|---|---|---|---|---|---|---|---|---|
| IPC | 1 | 10 | 50 | 500 | 1000 | 1 | 10 | 50 | 100 | 1 | 10 | 50 |
| Ratio | 0.02 | 0.2 | 1 | 10 | 20 | 0.2 | 2 | 10 | 20 | 0.2 | 2 | 10 |
| Random | 15.4±0.3 | 31.0±0.5 | 50.6±0.3 | 73.2±0.3 | 78.4±0.2 | 4.2±0.3 | 14.6±0.5 | 33.4±0.4 | 42.8±0.3 | 1.4±0.1 | 5.0±0.2 | 15.0±0.4 |
| KIP | 49.9±0.2 | 62.7±0.3 | 68.6±0.2 | - | - | 15.7±0.2 | 28.3±0.1 | - | - | 15.4±0.3 | 25.4±0.2 | - |
| FRePo | 46.8±0.7 | 65.5±0.4 | 71.7±0.2 | - | - | 28.7±0.1 | 42.5±0.2 | 44.3±0.2 | - | 15.4±0.3 | 25.4±0.2 | - |
| RCIG | 53.9±1.0 | 69.1±0.4 | 73.5±0.3 | - | - | 39.3±0.4 | 44.1±0.4 | 46.7±0.3 | - | 25.6±0.3 | 29.4±0.2 | - |
| DC | 28.3±0.5 | 44.9±0.5 | 53.9±0.5 | 72.1±0.4 | 76.6±0.3 | 12.8±0.3 | 25.2±0.3 | - | - | - | - | - |
| DM | 26.0±0.8 | 48.9±0.6 | 63.0±0.4 | 75.1±0.3 | 78.8±0.1 | 11.4±0.3 | 29.7±0.3 | 43.6±0.4 | - | 3.9±0.2 | 12.9±0.4 | 24.1±0.3 |
| DSA | 28.8±0.7 | 52.1±0.5 | 60.6±0.5 | 73.6±0.3 | 78.7±0.3 | 13.9±0.3 | 32.3±0.3 | 42.8±0.4 | - | - | - | - |
| TESLA | **48.5±0.8** | 66.4±0.8 | 72.6±0.7 | - | - | 24.8±0.4 | 41.7±0.3 | 47.9±0.3 | 49.2±0.4 | - | - | - |
| CAFE | 30.3±1.1 | 46.3±0.6 | 55.5±0.6 | - | - | 12.9±0.3 | 27.8±0.3 | 37.9±0.3 | - | - | - | - |
| MTT | 46.2±0.8 | 65.4±0.7 | 71.6±0.2 | - | - | 24.3±0.3 | 39.7±0.4 | 47.7±0.2 | 49.2±0.4 | 8.8±0.3 | 23.2±0.2 | 28.0±0.3 |
| FTD | 46.0±0.4 | 65.3±0.4 | 73.2±0.2 | - | - | 24.4±0.4 | 42.5±0.2 | 48.5±0.3 | 49.7±0.4 | 10.5±0.2 | 23.4±0.3 | 28.2±0.4 |
| ATT | 48.3±1.0 | 67.7±0.6 | 74.5±0.4 | - | - | 26.1±0.3 | 44.2±0.5 | 51.2±0.3 | - | 11.0±0.5 | 25.8±0.4 | - |
| DATM | 46.9±0.5 | 66.8±0.2 | 76.1±0.3 | 83.5±0.2 | 85.5±0.4 | 27.9±0.2 | 47.2±0.4 | 55.0±0.2 | 57.5±0.2 | 17.1±0.3 | 31.1±0.3 | 39.7±0.3 |
| **PAD** | 47.7±0.6 | **67.9±0.3** | **77.2±0.5** | **85.2±0.3** | **87.3±0.5** | **28.8±0.5** | **48.4±0.2** | **56.2±0.3** | **58.7±0.3** | **17.7±0.2** | **32.3±0.4** | **41.6±0.4** |
| Full Dataset | 84.8±0.1 | | | | | 56.2±0.3 | | | | 37.6±0.4 | | |

Table 1: Comparison with previous dataset distillation methods (bottom: matching-based, top: others) on CIFAR-10, CIFAR-100 and Tiny ImageNet. ConvNet is used for the distillation and evaluation. Our method consistently outperforms prior matching-based methods.

initial ratio (IR). Then, during training expert trajectories, samples are gradually added to the training set in order of increasing difficulty. After incorporating all the data into the training set, the scheduler will begin to remove easy samples from the target dataset. Unlike the gradual progression involved in adding data, the action of reducing data is completed in a single operation, since now the model has been trained on simple samples for a sufficient time. (Please refer to Appendix A.2 for experimental comparisons)

### 3.3 FILTERING INFORMATION EMBEDDING

To filter out misaligned information introduced by matching shallow-layer parameters, we propose to add a parameter selection module that masks out part of shallow layers for metric computation. Specifically, parameters of an agent network can be represented as a flattened array of length $L$ that stores weights of agent models ordered from shallow to deep layers (parameters within the same layer are sorted in default order). The parameter selection sets a threshold ratio $\alpha$ such that the first $k = L \cdot \alpha$ parameters are not used for distillation. Then the parameters used for matching can now be formulated as:

$$\hat{\theta}_{t+N} = \{\underbrace{\hat{\theta}_0, \hat{\theta}_1, \cdots, \hat{\theta}_{k-1}}_{\text{discard}}, \underbrace{\hat{\theta}_k, \hat{\theta}_{k+1}, \cdots, \hat{\theta}_L}_{\text{used for matching}}\}. \tag{4}$$

In practice, the ratio $\alpha$ should vary with the change of IPC. For smaller IPCs, it is necessary to incorporate basic information thus $\alpha$ should be lower. Conversely, basic information is redundant in larger IPC cases, so $\alpha$ should be higher accordingly.

## 4 EXPERIMENTS

### 4.1 SETTINGS

We compare PAD with several prominent dataset distillation methods, which can be divided into two categories: matching-based approaches including DC (Zhao et al., 2020), DM (Zhao & Bilen, 2021a), DSA (Zhao & Bilen, 2021b), CAFE (Wang et al., 2022), MTT (Cazenavette et al., 2022), FTD (Du et al., 2022), ATT (Liu et al., 2024), DATM (Guo et al., 2023), TESLA (Cui et al., 2022), and kernel-based approaches including KIP (Nguyen et al., 2020), FRePo (Zhou et al., 2022), RCIG (Loo et al., 2023). The assessment is conducted on widely recognized datasets: CIFAR-10, CIFAR-100(Krizhevsky, 2009), and TinyImageNet (Le & Yang, 2015). We implemented our method based on DATM (Guo et al., 2023). In both the distillation and evaluation phases, we apply the standard set of differentiable augmentations commonly used in previous studies (Cazenavette et al., 2022; Du et al., 2022; Guo et al., 2023). By default, networks are constructed with instance normalization unless explicitly labeled with "-BN," indicating batch normalization (e.g., ConvNet-BN). For CIFAR-10 and CIFAR-100, distillation is typically performed using a 3-layer ConvNet, while Tiny ImageNet requires a 4-layer ConvNet. Cross-architecture experiments also utilize LeNet (LeCun et al., 1998),

| Dataset | Ratio | Method | ConvNet | ConvNet-BN | ResNet18 | ResNet18-BN | VGG11 | AlexNet | LeNet | MLP | Avg. |
|---|---|---|---|---|---|---|---|---|---|---|---|
| CIFAR-10 | 20% | Random | 78.38 | 80.25 | 84.58 | 87.21 | 80.81 | 80.75 | 61.85 | 50.98 | 75.60 |
| | | Glister | 62.46 | 70.52 | 81.10 | 74.59 | 78.07 | 70.55 | 56.56 | 40.59 | 66.81 |
| | | Forgetting | 76.27 | 80.06 | 85.67 | 87.18 | 82.04 | 81.35 | 64.59 | 52.21 | 76.17 |
| | | DATM | 85.50 | 85.23 | **87.22** | **88.13** | **84.65** | 85.14 | 66.70 | 52.40 | 79.37 |
| | | **PAD** | **87.25** | **85.67** | 86.95 | 88.09 | 84.34 | **85.83** | **67.28** | **53.62** | **79.84** |
| | | ↑ | +8.87 | +5.42 | +2.37 | +0.88 | +3.53 | +5.08 | +5.43 | +2.64 | +4.28 |
| CIFAR-100 | 20% | Random | 42.80 | 46.38 | 47.48 | 55.62 | 42.69 | 38.05 | 25.91 | 20.66 | 39.95 |
| | | Glister | 35.45 | 37.13 | 42.49 | 46.14 | 43.06 | 28.58 | 23.33 | 17.08 | 34.16 |
| | | Forgetting | 45.52 | 49.99 | 51.44 | 54.65 | 43.28 | 43.47 | 27.22 | 22.90 | 42.30 |
| | | DATM | 57.50 | 57.75 | 57.98 | **63.34** | **55.10** | 55.69 | 33.57 | 26.39 | 50.92 |
| | | **PAD** | **58.71** | **58.66** | **58.15** | 63.17 | 55.02 | **55.93** | **33.87** | **27.12** | **51.30** |
| | | ↑ | +15.91 | +12.28 | +10.67 | +7.55 | +12.33 | +17.88 | +7.96 | +6.46 | +11.36 |
| TinyImageNet | 10% | Random | 15.00 | 24.21 | 17.73 | 28.07 | 22.51 | 14.03 | 9.25 | 5.85 | 17.08 |
| | | Glister | 17.32 | 19.77 | 18.84 | 23.12 | 19.10 | 11.68 | 8.84 | 3.86 | 15.32 |
| | | Forgetting | 20.04 | 23.83 | 19.38 | 28.88 | 23.77 | 12.13 | 12.06 | 5.54 | 18.20 |
| | | DATM | 39.68 | 40.32 | **36.12** | **43.14** | 38.35 | **35.10** | 12.41 | 9.02 | 31.76 |
| | | **PAD** | **41.55** | **40.88** | 36.08 | 42.96 | **38.64** | 35.02 | **13.17** | **9.68** | **32.18** |
| | | ↑ | +26.55 | +16.67 | +18.35 | +14.89 | +16.13 | +20.99 | +3.92 | +3.83 | +15.18 |

Table 2: Cross-architecture evaluation of distilled data on unseen networks. Results worse than random selection are indicated with red color. ↑ denotes the performance improvement brought by our method compared with random selection. Tiny denotes Tiny ImageNet.

AlexNet (Krizhevsky et al., 2012), VGG11 (Simonyan & Zisserman, 2014), and ResNet18 (He et al., 2015). More details can be found in the appendix.

## 4.2 MAIN RESULTS

**CIFAR and Tiny ImageNet** We conduct comprehensive experiments to compare the performance of our method with previous works. As the results presented in Table 1, PAD outperforms previous matching-based methods on three datasets except for the case when IPC=1. When compared with kernel-based methods, which use a larger network to perform the distillation, our technique exhibits superior performance in most cases, particularly when the compression ratio exceeds $1\%$. As can be observed, PAD performs relatively better when IPC is high, enabling the setting of IPC500 on CIFAR-10 to also achieve lossless performance. This suggests that our filtering out misaligned information strategy becomes increasingly effective as IPC increases. More comparisons can be found in Appendix A.3

**Cross Architecture Generalization** We evaluate the generalizability of our distilled data in both low and high IPC cases. As reflected in Table 2, our distilled datasets on large IPCs also have the best performance on most evaluated architectures, showing good generalizability in the low compressing ratio case. Moreover, as results reported in Table 3a, when IPC is small, our distilled data outperforms the previous SOTA method DATM on ResNet and AlexNet while maintaining comparable accuracy on VGG. This suggests that our distilled data on high compressing ratios generalizes well across various unseen networks.

## 4.3 ABLATION STUDY

To validate the effectiveness of each component of our method, we conducted ablation experiments on modules (section 4.3.1) and their hyper-parameter settings (section 4.3.2 and section 4.3.2). For the results below, we only report the mean performance of multiple runs.

### 4.3.1 MODULES

Our method incorporates two separate modules to filter information extraction (FIEX) and information embedding (FIEM), respectively. To verify their isolated effectiveness, we conduct an ablation study by applying two modules individually. As depicted in Table 3b, both FIEX and FIEM bring improvements, implying their efficacy. By applying these two modules, we are able to effectively remove unaligned information, improving the distillation performance.

| Method | ConvNet | ResNet18 | VGG | AlexNet |
|--------|---------|----------|-----|---------|
| Random | 33.5 | 32.0 | 32.2 | 26.7 |
| FTD | 48.9 | 46.7 | 43.2 | 42.2 |
| DATM | 55.0 | 51.7 | **45.4** | 45.7 |
| **PAD** | **56.2** | **52.4** | 45.0 | **45.9** |

| FIEX | FIEM | Accuracy(%) |
|------|------|-------------|
| | | 66.7 |
| | ✓ | 67.3 |
| ✓ | | 67.6 |
| ✓ | ✓ | 67.9 |

| FIEX | FIEM | Accuracy(%) |
|------|------|-------------|
| | | 55.0 |
| | ✓ | 55.5 |
| ✓ | | 55.8 |
| ✓ | ✓ | 56.2 |

(a) Datasets distilled by PAD generalize well across various architectures.

(b) Both FIEX and FIEM bring non-trivial improvements to the baseline.

Table 3: **(a)** Cross-Architecture evaluation on CIFAR-100 IPC50. **(b)** Ablation studies on the modules of our method on CIFAR-10 IPC10 and CIFAR-100 IPC50.

| IR | AEE | | |
|----|-----|-----|-----|
| | 20 | 40 | 60 |
| 50% | 66.2 | 66.1 | 65.9 |
| 75% | **67.8** | **67.5** | **66.6** |
| 80% | 67.6 | 67.4 | 66.5 |

| Method | IPC | | |
|--------|-----|-----|-----|
| | 1 | 10 | 500 |
| Loss | 45.7 | 66.5 | 83.5 |
| Uncertainty | 46.2 | 67.0 | 84.2 |
| EL2N | **47.7** | **67.9** | **85.2** |

| IPC | Ratio | | | |
|-----|-----|-----|-----|-----|
| | 0% | 25% | 50% | 75% |
| 1 | **47.7** | 46.6 | 46.0 | 41.3 |
| 10 | 67.2 | **67.7** | 66.9 | 65.2 |
| 500 | 83.7 | 83.8 | **85.2** | 84.6 |

(a) Set IR as 75% always performs best on different add-end-epochs.

(b) Using EL2N to measure the difficulty of samples has the best performance.

(c) As IPC increases, removing more shallow-layer parameters becomes more effective.

Table 4: **(a)** Ablation of the initial ratio for the trajectory training on CIFAR-10 IPC10. **(b)** Ablation of different difficulty scoring functions on CIFAR-10. **(c)** Results of masking out different ratios of shallow-layer parameters across various IPCs on CIFAR-10.

### 4.3.2 HYPER-PARAMETERS OF FILTERING INFORMATION EXTRACTION

**Initial Ratio and Data Addition Epoch**    To filter the information learned by agent models, we initialize the training set with only easy samples, and the size is determined by a certain ratio of the total size. Then, we gradually add hard samples into the training set. In practice, we use two hyper-parameters to control the addition process: the initial ratio (IR) of training data for training set initialization and the end epoch of hard sample addition (AEE). These two parameters together control the amount of data agent models can see at each epoch and the speed of adding hard samples.

In Table 4a, we show the distillation results where different hyper-parameters are utilized. In general, a larger initial ratio and faster speed of addition bring better performances. Although the distillation benefited more from learning simpler information when IPC is small (Guo et al., 2023), our findings indicate that excessively removing difficult samples (e.g., more than a quarter) early in the training phase can adversely affect the distilled data. This negative impact is likely due to the excessive removal, which leads to distorted feature distributions within each category. On the other hand, reasonably improving the speed of adding hard samples allows the agent model to achieve a more balanced learning of information of varying difficulty across different stages.

**Other Difficulty Scoring Functions**    Identifying the difficulty of data points is the key to filtering out misaligned information in the extraction step. Here, we compare the effect of using other difficulty-scoring functions to evaluate the difficulty of data. (1) prediction loss of a pre-trained ResNet. (2) uncertainty score (Coleman et al., 2019). (3) EL2N (Paul et al., 2021). As can be observed in Table 4b, EL2N performs the best across various IPCs; thus, we use it to measure how hard each data point is as default in our method. Note that this can also be replaced with a more advanced data selection algorithm.

### 4.3.3 RATIOS OF PARAMETER SELECTION

It is important to find a good balance between the percentage of shallow-layer parameters removed from matching and the loss of information. In Table 4c, we show results obtained on different IPCs by discarding various ratios of shallow-layer parameters. The impact of removing varying proportions of shallow parameters on the distilled data and its relationship with changes in IPC is consistent with prior conclusions. For small IPCs, distilled data requires more low-level basic information. Thus, removing too many shallow-layer parameters causes a negative effect on the classification performance. By contrast, high-level semantic information is more important when it comes to

| IPC | 25% | 50% | 75% | baseline |
|---|---|---|---|---|
| 1 | 44.1 | 43.2 | 41.8 | 46.9 |
| 10 | 62.2 | 57.7 | 51.1 | 66.9 |
| 50 | 69.2 | 66.5 | 58.3 | 76.1 |

(a) Dicarding deep-layer parameters significantly harms the performance.

| IPC | PAD | BLiP |
|---|---|---|
| 1 | 46.8 (**+0.6**, 80%) | 46.3 (+0.2, 80%) |
| 10 | 66.5 (**+1.1**, 90%) | 65.7 (+0.4, 90%) |
| 50 | 73.0 (**+1.4**, 95%) | 72.0 (+0.4, 95%) |

(b) Data selection (FIEX) in PAD is more effective in improving trajectory matching.

| IPC | $SRe^2L$ | $SRe^2L$ + PAD |
|---|---|---|
| 1 | 25.4 | **26.7** (↑ 1.3) |
| 10 | 28.2 | **29.3** (↑ 1.1) |
| 50 | 57.2 | **57.9** (↑ 0.7) |

(c) PAD can also be applied to $SRe^2L$ and brings non-trivial improvements.

Table 5: (a) Ablation results of discarding deep-layer parameters during information embedding on CIFAR-10. (b) We compare our data selection strategy with that of BLiP on CIFAR10. The left in the bracket denotes the improvement over MTT, and the right denotes the percentage of real data used for distillation. (c) Results of $SRe^2L$ on CIFAR-100 after applying PAD.

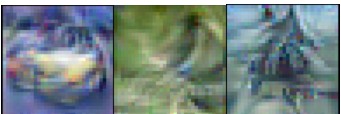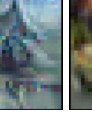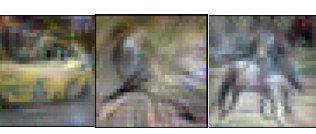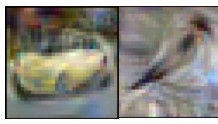

(a) with 100% parameters        (b) with 75% parameters        (c) with 50% parameters

Figure 4: Synthetic images of CIFAR-10 IPC50 obtained by PAD with different ratios of parameter selection. Smoother image features indicate that by removing some shallow-layer parameters during matching, PAD successfully filters out coarse-grained low-level information.

large IPCs. With increasing ratios of shallow-layer parameters being discarded, we can ensure that low-level information is effectively filtered out from the distilled data.

## 5 DISCUSSION

### 5.1 DISTILLED IMAGES WITH FILTERING INFORMATION EMBEDDING

To see the concrete patterns brought by removing shallow-layer parameters to perform the trajectory matching, we present distilled images obtained by discarding various ratios of shallow-layer parameters in Figure 4. As can be observed in Figure 4a, without removing any shallow-layer parameters to filter misaligned information, synthetic images are interspersed with substantial noises. These noises often take the form of coarse and generic information, such as the overall color distribution and edges in the image, which provides minimal utility for precise classification.

By contrast, images distilled by our enhanced methodology (see Figure 4b and Figure 4c), which includes meticulous masking out shallow-layer parameters during trajectory matching according to the compressing ratio, contain more fine-grained and smoother features. These images also encapsulate a broader range of semantic information, which is crucial for helping the model make accurate classifications. Moreover, we observe a clear trend: as the amount of the removed shallow-layer parameters increases, the distilled images exhibit clearer and smoother features.

### 5.2 RATIONALE FOR PARAMETER SELECTION

In this section, we analyze why shallow-layer parameters should be masked out from the perspective of trajectory matching. In Figure 5, we present the changes in trajectory matching loss across different layers as the distillation progresses. Compared to the deep-layer parameters of the agent model, a substantial number of shallow-layer parameters exhibit low loss values that fluctuate during the matching. By contrast, losses of the deep layers are much higher but consistently decrease as distillation continues. This suggests that matching shallow layers primarily conveys low-level information that is readily captured by the synthetic data and quickly saturated. Thus, the excessive addition of such low-level information produces noise, reducing the quality of distilled datasets.

For a concrete visualization, we provide distilled images resulting from using only shallow-layer parameters or only deep-layer parameters to match trajectories in Figure 6. The coarse image features depicted in Figure 6a further substantiate our analysis.

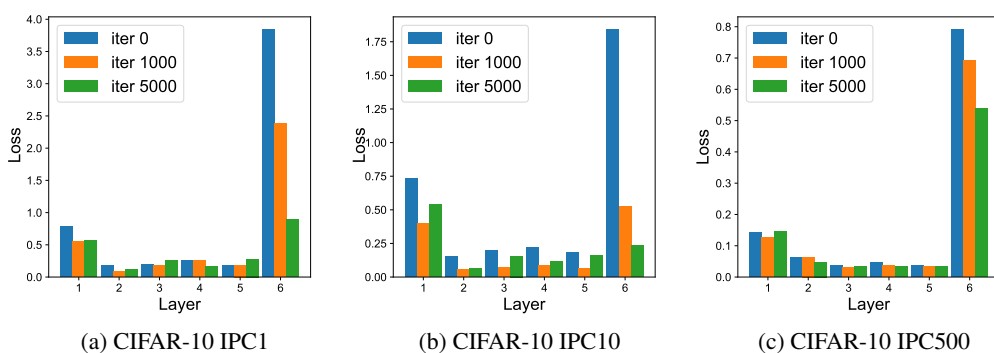

(a) CIFAR-10 IPC1      (b) CIFAR-10 IPC10      (c) CIFAR-10 IPC500

Figure 5: Losses of different layers of ConvNet after matching trajectories for 0, 1000, and 5000 iterations. We notice a similar phenomenon on both small (IPC1 and IPC10) and large IPCs (IPC500): losses of shallow-layer parameters fluctuate along the matching process, while losses of deep-layer parameters show a clear trend of decreasing.

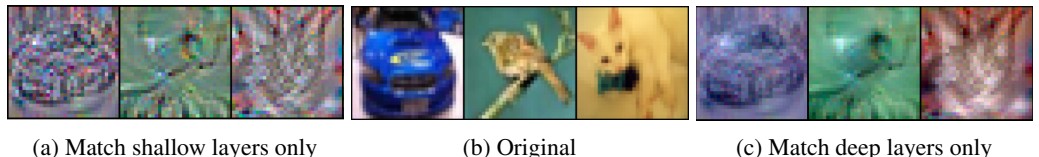

(a) Match shallow layers only      (b) Original      (c) Match deep layers only

Figure 6: Synthetic images visualization with parameter selection. Matching parameters in shallow layers produces an abundance of low-level texture features, whereas patterns generated by matching deep-layer parameters embody richer high-level semantic information.

To further demonstrate the importance of deep-layer parameters, we show performances of discarding deep-layer parameters in Table 5a. As can be observed, there are significant performance drops when these parameters are not used for distillation. As the discarding ratio increases, the performance drop becomes more serious for all IPCs. Also, the impact of discarding deep-layer parameters is more significant on larger IPCs. These results verify that deep-layer parameters are more important than shallow-layer parameters.

### 5.3 OTHER METHODS WITH DATA SELECTION

To further demonstrate the effectiveness of our FIEX, we compare ours with BLiP (Xu et al., 2023), which also uses a data selection strategy before distillation. It proposes a data utility indicator to evaluate if samples are 'useful' given an IPC setting, and then samples with low utility are pruned. As shown in Table 5b, PAD brings better performance improvements on IPC1/10/50. Under a given data-dropping ratio, PAD's improvements over BLiP get larger as the IPC increases. This supports our conclusion that difficulty misalignment between IPCs and real data used is more harmful. PAD's data selection module is more effective in removing such misaligned information.

### 5.4 GENERALIZATION TO RECENT ADVANCEMENTS

In Section 2, we show that the two filtering modules of PAD can be applied on seminal matching-based DD baselines (DC, DM, MTT) and improve their performances remarkably. To catch up with the latest DD progress, we combine PAD with a more recent DD method that achieves great success on high-resolution datasets, $SRe^2L$ (Yin et al., 2023), to show that PAD can also generalize well on other methods. As shown in Table 5c, by filtering out misaligned information extracted in $SRe^2L$'s *squeeze* and *recover* stages, the performance of $SRe^2L$ improves on both small and large IPC settings. Particularly, PAD's information extraction filter brings more pronounced improvements to smaller IPCs. This further validates PAD's efficacy in aligning information of dataset distillation and

demonstrates that PAD also has decent generalizability on other methods that involve an information-extraction-like or information-embedding-like component.

# 6    RELATED WORK

Introduced by Wang et al. (2020), dataset distillation aims to synthesize a compact set of data that allows models to achieve similar test performances compared with the original dataset. Since then, a number of studies have explored various approaches. Most of the popular methods can be divided into three types: matching-based, generative-model-based, and knowledge-distillation-based.

**Matching-based methods.** These methods first use agent models to extract information from the target dataset by recording a specific metric (Du et al., 2023; Lee et al., 2022; Shin et al., 2023; Liu et al., 2023). Representative works that design different metrics include DC (Zhao et al., 2020) that matches gradients, DM (Zhao & Bilen, 2021a) that matches distributions, and MTT (Cazenavette et al., 2022) that matches training trajectories. Then, the distilled dataset is optimized by minimizing the matched distance between the metric computed on synthetic data and the record one from the previous step. Following this workflow, many works improved the efficacy of the distilled dataset.

For example, CAFE (Wang et al., 2022) preserves the real feature distribution and the discriminative power of the synthetic data and achieves prominent generalization ability across various architectures. DATM (Guo et al., 2023) proposes to match early trajectories for small IPCs and late trajectories for large IPCs, achieving SOTA performances on several benchmarks. BLiP (Xu et al., 2023) discovers the issue of data redundancy in the previous distillation framework and propose to prune the real dataset before distillation. PDD (Chen et al., 2023) identifies the change of learned pattern complexity at different training stages and proposes a multi-stage distillation process where each synthetic subset is conditioned on the previous ones to alleviate the above challenge. Moreover, new metrics such as spatial attention maps (Sajedi et al., 2023; Khaki et al., 2024) have also been introduced and achieved promising performance in distilling large-scale datasets. Despite these advancements, matching-based methods often overlook the misalignment in information extraction and information embedding, restricting their performances to be further improved. **Generative-model based methods.** GANs (Goodfellow et al., 2014; Karras et al., 2018; 2019; Wang et al., 2023) and diffusion models (Rombach et al., 2021; Moser et al., 2024; Gu et al., 2023) can also be used to synthesize high quality datasets. DiM (Wang et al., 2023) uses deep generative models to store information of the target dataset. GLaD (Cazenavette et al., 2023) transfers synthetic data optimization from the pixel space to the latent space by employing deep generative priors. It enhances the generalizability of previous methods.

**Knowledge-distillation-based methods.** Different from previous dataset distillation approaches, methods following this track apply knowledge distillation during the evaluation of synthetic data. $SRe^2L$ (Yin et al., 2023) introduces a "squeeze, recover, relabel" procedure that decouples previous bi-level optimization and achieves success on high-resolution settings with lower computational costs. RDED (Sun et al., 2023) proposes a computationally efficient DD method that doesn't require synthetic image optimization by extracting and rearranging key image patches.

# 7    CONCLUSION

In this work, we find a limitation of existing Dataset Distillation methods in that they will introduce misaligned information to the distilled datasets. To alleviate this, we propose PAD, which incorporates two modules to filter out misaligned information. For information extraction, PAD prunes the target dataset based on sample difficulty for different IPCs so that only information with aligned difficulty is extracted by the agent model. For information embedding, PAD discards part of shallow-layer parameters to avoid injecting low-level basic information into the synthetic data. PAD achieves SOTA performance on various benchmarks. Moreover, we show PAD can also be applied to methods based on matching gradients and distributions, bringing improvements across various IPC settings.

**Limitations**    Our alignment strategy could also be applied to methods based on matching gradients and distributions. However, due to the limitation of computing resources, for methods based on matching distributions and gradients, we have only validated our method's effectiveness on DM (Zhao & Bilen, 2021a) and DC (Zhao et al., 2020) (see Table 6 and Table 7).

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

| IPC | Ratio | | | | | | | Baseline |
|-----|-------|-----|-----|-----|-----|-----|-----|----------|
|     | 5% | 10% | 15% | 20% | 25% | 30% | 50% | |
| 1 | 28.0 | 28.4 | 28.5 | **29.1** | 28.8 | 28.1 | 27.9 | 27.8 |
| 10 | 45.2 | 45.5 | 45.7 | 46.1 | **46.3** | 45.3 | 44.5 | 44.7 |
| 500 | 71.7 | **71.9** | 71.2 | 71.4 | 70.3 | 69.8 | 67.1 | 71.4 |

(a) Removing various ratios of hard/easy samples improves DC on small/large IPCs.

| IPC | Ratio | | | | | | | Baseline |
|-----|-------|-----|-----|-----|-----|-----|-----|----------|
|     | 5% | 10% | 15% | 20% | 25% | 30% | 50% | |
| 1 | 26.8 | 27.1 | 27.3 | 27.9 | 28.2 | 28.5 | **29.2** | 26.4 |
| 10 | 48.6 | 48.9 | 49.7 | **50.3** | 49.6 | 49.2 | 48.5 | 48.4 |
| 500 | 75.6 | 76.2 | **76.3** | 75.8 | 75.3 | 74.6 | 74.2 | 75.1 |

(b) Removing various ratios of hard/easy samples improves DM on small/large IPCs.

Table 6: Results of filtering information extraction by removing hard/easy samples in DC(a) and DM(b) on CIFAR-10.

# APPENDIX

## A  ADDITIONAL EXPERIMENTAL RESULTS AND FINDINGS

### A.1  FILTERING MISALIGNED INFORMATION IN DC AND DM

Although PAD is implemented based on trajectory matching methods, we also test our proposed data alignment and parameter alignment on gradient matching and distribution matching. The performances of enhanced DC and DM with each of the two modules are reported in Table 6 and Tabl 7, respectively. We provide details of how we integrate these two modules into gradient matching and distribution matching in the following sections.

**Gradient Matching** We use the official implementation[1] of DC (Zhao et al., 2020). In the Information Extraction step, DC uses an agent model to calculate the gradients after being trained on the target dataset. We employ filter misaligned information in this step as follows: When IPC is small, a certain ratio of hard samples is removed from the target dataset so that the recorded gradients only contain simple information. Conversely, when IPC becomes large, we remove easy samples instead.

In the Information Embedding step, DC optimizes the synthetic data by back-propagating on the gradient matching loss. The loss is computed by summing the differences in gradients between each pair of model parameters. Thus, we apply parameter selection by discarding a certain ratio of parameters in the shallow layers.

**Distribution Matching** We use the official implementation of DM (Zhao & Bilen, 2021a), which can be accessed via the same link as DC. In the Information Extraction step, DM uses an agent model to generate embeddings of input images from the target dataset. Similarly, filtering information extraction is applied by removing hard samples for small IPCs and easy samples for large IPCs.

In the Information Embedding step, since DM only uses the output of the last layer to match distributions, we modify the implementation of the network such that outputs of each layer in the model are returned by the forward function. Then, we perform parameter selection following the same practice as before.

### A.2  DATA SCHEDULER

To support the way we design the data scheduler to remove easy samples at late trajectories directly, we compare direct removal with gradual removal. The implementation of gradual removal is similar to the hard sample addition. Experimental results are shown in Table 8(8a) on CIFAR-10 and CIFAR-100. Only large IPCs are tested because only large IPCs match late trajectories. As can be observed, compared with gradually removing easy data, deleting easy samples in one operation performs better. This supports our conclusion that after being trained on the full dataset for some

---
[1]https://github.com/VICO-UoE/DatasetCondensation.git

| IPC | Ratio | | | Baseline |
|---|---|---|---|---|
| | 25% | 50% | 75% | |
| 10 | **45.2** | 44.7 | 43.8 | 44.9 |
| 500 | 72.5 | 72.8 | **73.4** | 72.2 |

(a) Matching gradients from deep-layer parameters leads to improvements.

| IPC | Ratio | | | Baseline |
|---|---|---|---|---|
| | 25% | 50% | 75% | |
| 10 | **49.5** | 49.1 | 48.3 | 48.9 |
| 500 | 75.5 | 75.9 | **76.3** | 75.1 |

(b) Matching distributions from deep-layer parameters leads to improvements.

Table 7: Results of filtering information embedding by masking out shallow-layer parameters for metric computation in DC(a) and DM(b) on CIFAR-10.

| Strategy | CIFAR-10 IPC | | CIFAR-100 IPC | |
|---|---|---|---|---|
| | 50 | 100 | 50 | 100 |
| Gradually remove | 84.2 | 86.4 | 55.6 | 58.3 |
| Directly remove | **84.6** | **86.7** | **55.9** | **58.5** |

(a) Directly removing easy samples at late trajectories brings better performances.

| FIEX | FIEM | Accuracy(%) |
|---|---|---|
| | | 55.0 |
| | ✓ | 55.5 |
| ✓ | | 55.8 |
| ✓ | ✓ | 56.2 |

(b) Each module brings non-trivial improvements to the baseline.

Table 8: (a) Comparison between gradually removing easy samples and directly removing easy samples during trajectory training. (b) Ablation results on CIFAR-100 IPC50.

epochs, it is more effective for the model to focus on learning hard information rather than easy information by removing easy samples directly.

A.3 COMPARISON WITH RECENT ADVANCES

Although PAD is designed to filter out misaligned information in matching-based methods, we are aware of the recent advancements in the DD field. Therefore, we compare two more recent methods, RDED (Sun et al., 2023) and SPEED (Wei et al., 2023).

**RDED** is a non-optimization-based method that selects important image tokens and reconstructs them back to images. During evaluation, knowledge distillation is applied by minimizing the Kullback-Leibler (KL) divergence between the student model's output and the teacher model's output on the same batch of synthetic data. To compare with such a method, we adopt its knowledge distillation strategy during evaluation to ensure a fair comparison. Other experimental settings of PAD are the same. As shown in Table 9, PAD demonstrates superior performances in most of the settings, especially when the compressing ratio is higher (larger IPCs). On small IPCs, the knowledge distillation leads to a drop in PAD's pure DD performance (Table 1). This is because knowledge from a well-trained teacher exceeds the capacity of small IPCs.

**SPEED** is a parameterization-based method that prioritizes proper parameterization of the synthetic dataset. It proposes spatial-agnostic epitomic tokens and sparse coding matrices to reduce spatial redundancy. In Table 10, we show that although SPEED excels at IPC1, it is difficult for the method to scale to higher compressing ratios effectively. By contrast, PAD shows prominent scalability on IPC10 and IPC50 with non-trivial performance improvements. In terms of computational efficiency, both methods use trajectory-matching as the backbone, but SPEED requires more optimizations on the feature-recurrent network.

A.4 INSENSITIVITY TO HYPER-PARAMETERS

Although PAD introduces two hyper-parameters in the FIEX module, its performance is not sensitive to their values, and they are easy to tune. In Table 11, we show different combinations of AEE and IR on three datasets. We find that when the AEE is 20 or 40, and IR is around 75% to 80%, the change in performance is marginal. This shows that these two hyper-parameters do not significantly influence PAD's performance within a proper range.

| Dataset | CIFAR10 | | | CIFAR-100 | | | TinyImageNet | | |
|---------|-----|-----|-----|-----|-----|-----|-----|-----|-----|
| IPC | 1 | 10 | 50 | 1 | 10 | 50 | 1 | 10 | 50 |
| RDED | 23.5±0.3 | 50.2±0.3 | 68.4±0.1 | **19.6±0.3** | 48.1±0.3 | 57.0±0.1 | **12.0±0.1** | **39.6±0.1** | 47.6±0.2 |
| PAD | **26.8±0.6** | **62.3±0.5** | **78.5±0.3** | 16.4±0.4 | **50.2±0.3** | **59.3±.2** | 10.1±0.3 | 37.2±0.5 | **48.5±0.3** |

Table 9: Comparison between PAD and RDED on CIFAR-10, CIFAR-100, and TinyImageNet. PAD achieves superior performances in 6 out of 9 settings. On larger IPCs, PAD's advantage is more pronounced.

| Dataset | CIFAR-100 | | | TinyImageNet | | |
|---------|-----|-----|-----|-----|-----|-----|
| IPC | 1 | 10 | 50 | 1 | 10 | 50 |
| SPEED | **40.4±0.4** | 45.9±0.3 | 49.1±0.2 | **26.9±0.3** | 28.8±0.2 | 30.1±0.3 |
| PAD | 16.4±0.4 | **50.2±0.3** | **59.3±.2** | 10.1±0.3 | **37.2±0.5** | **48.5±0.3** |

Table 10: Comparison between PAD and SPEED on CIFAR-100 and TinyImageNet. PAD demonstrates superior scalability on larger IPCs.

## B  EXPERIMENTAL SETTINGS

We use DATM (Guo et al., 2023) as the backbone TM algorithm, and our proposed PAD is built upon. Thus, our configurations for distillation, evaluation, and network are consistent with DATM.

**Distillation.** We conduct the distillation process for 10,000 iterations to ensure full convergence of the optimization. By default, ZCA whitening is applied in all the experiments.

**Evaluation.** We train a randomly initialized network on the distilled dataset and evaluate its performance on the entire validation set of the original dataset. Following DATM (Guo et al., 2023), the evaluation networks are trained for 1000 epochs to ensure full optimization convergence. For fairness, the experimental results of previous distillation methods in both low and high IPC settings are sourced from (Guo et al., 2023).

**Network.** We employ a range of networks to assess the generalizability of our distilled datasets. For scaling ResNet, LeNet, and AlexNet to Tiny-ImageNet, we modify the stride of their initial convolutional layer from 1 to 2. In the case of VGG, we adjust the stride of its final max pooling layer from 1 to 2. The MLP used in our evaluations features a single hidden layer with 128 units.

**Hyper-parameters.** Hyper-parameters of our experiments on CIFAR-10, CIFAR-100, and TinyImageNet are reported in Table 12. Hyper-parameters can be divided into three parts, including FIEX, FIEM, and trajectory matching (TM). For FIEX, the ratio of easy samples removed for all IPCs is 10%. Soft labels are applied in all experiments, we set its momentum to 0.9.

**Compute resources.** Our experiments are run on 4 NVIDIA A100 GPUs, each with 80 GB of memory. The amount of GPU memory needed is mainly determined by the batch size of synthetic data and the number of steps that the agment model is trained on synthetic data. To reduce the GPU usage when IPC is large, one can apply TESLA (Cui et al., 2022) or simply reducing the synthetic steps $N$ or the synthetic batch size. However, the decrement of hyper-parameters shown in Table 12 could result in performance degradation.

| IR | AEE | | |
|---|---|---|---|
| | 20 | 30 | 40 |
| 50% | 66.2 | 65.9 | 66.1 |
| 75% | **67.8** | 67.6 | **67.5** |
| 80% | 67.6 | **67.7** | 67.4 |

(a) CIFAR-10 IPC10

| IR | AEE | | |
|---|---|---|---|
| | 20 | 30 | 40 |
| 50% | 47.8 | 47.7 | 47.7 |
| 75% | **48.3** | 48.1 | **48.4** |
| 80% | 48.2 | **48.3** | 48.3 |

(b) CIFAR-100 IPC10

| IR | AEE | | |
|---|---|---|---|
| | 20 | 30 | 40 |
| 50% | 16.9 | 16.8 | 17.3 |
| 75% | **17.4** | **17.5** | **17.7** |
| 80% | 17.3 | 17.4 | 17.6 |

(c) TinyImageNet IPC1

Table 11: Setting IR=75% and AEE=40 generalize well across various datasets. Generally, 75%-80% for IR and 20-40 for AEE are good settings with minor performance fluctuation.

| Dataset | IPC | DA | | PA | | | | TM | | | | | |
|---|---|---|---|---|---|---|---|---|---|---|---|---|---|
| | | IR | AEE | $\alpha$ | N | M | $T^-$ | $T$ | $T^+$ | Interval | Synthetic Batch Size | Learning Rate (Label) | Learning Rate (Pixels) |
| CIFAR-10 | 1 | 0.75 | 20 | 0% | 80 | 2 | 0 | 4 | 4 | - | 10 | 5 | 100 |
| | 10 | | | 25% | 80 | 2 | 0 | 10 | 20 | 100 | 100 | 2 | 100 |
| | 50 | | | 25% | 80 | 2 | 0 | 20 | 40 | 100 | 500 | 2 | 1000 |
| | 500 | | | 50% | 80 | 2 | 40 | 60 | 60 | - | 1000 | 10 | 50 |
| | 1000 | | | 75% | 80 | 2 | 40 | 60 | 60 | - | 1000 | 10 | 50 |
| CIFAR-100 | 1 | 0.75 | 40 | 0% | 40 | 3 | 0 | 10 | 20 | 100 | 100 | 10 | 1000 |
| | 10 | | | 25% | 80 | 2 | 0 | 20 | 40 | 100 | 1000 | 10 | 1000 |
| | 50 | | | 50% | 80 | 2 | 40 | 60 | 80 | 100 | 1000 | 10 | 1000 |
| | 100 | | | 50% | 80 | 2 | 40 | 80 | 80 | - | 1000 | 10 | 50 |
| TI | 1 | 0.75 | 40 | 0% | 60 | 2 | 0 | 15 | 30 | 400 | 200 | 10 | 10000 |
| | 10 | | | 25% | 60 | 2 | 0 | 20 | 40 | 100 | 250 | 10 | 100 |
| | 50 | | | 50% | 80 | 2 | 20 | 40 | 60 | 100 | 250 | 10 | 100 |

Table 12: Hyper-parameters for different benchmarks.

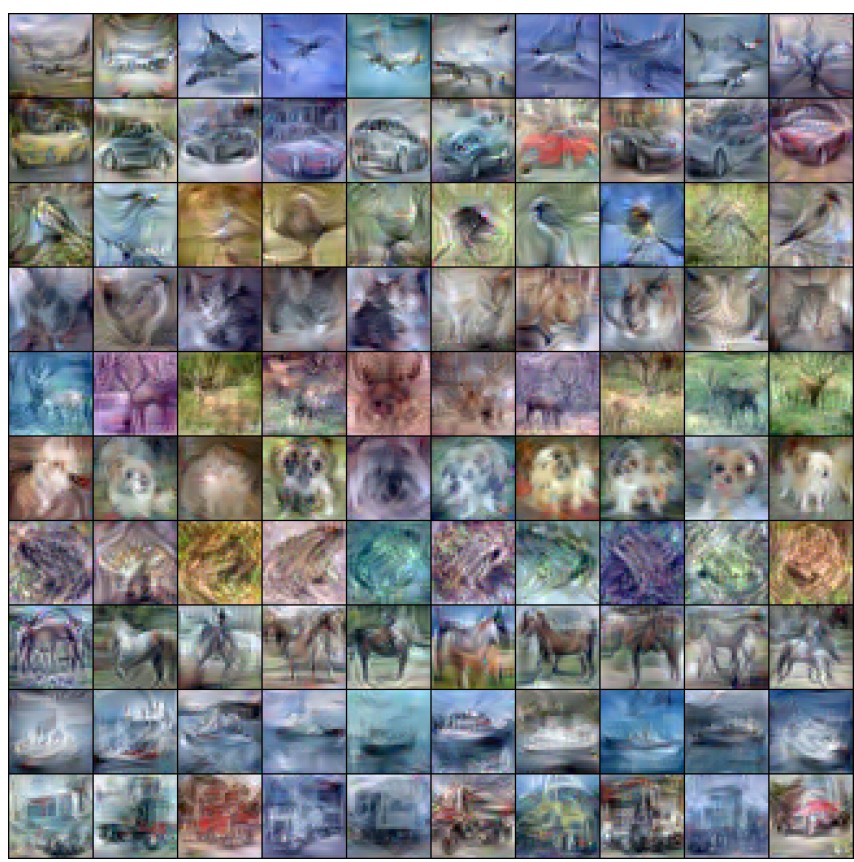

Figure 7: Distilled images of CIFAR-10 IPC10

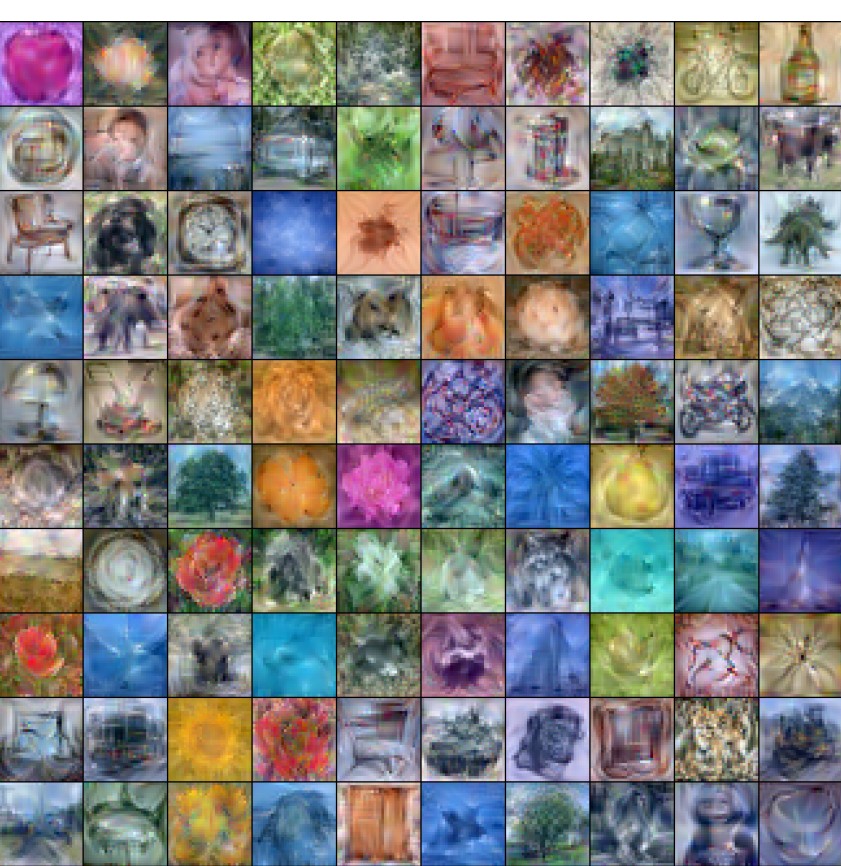

Figure 8: Distilled images of CIFAR-100 IPC1

972
973
974
975
976
977
978
979
980
981
982
983
984
985
986
987
988
989
990
991
992
993
994
995
996
997
998
999
1000
1001
1002
1003
1004
1005
1006
1007
1008
1009
1010
1011
1012
1013
1014
1015
1016
1017
1018
1019
1020
1021
1022
1023
1024
1025

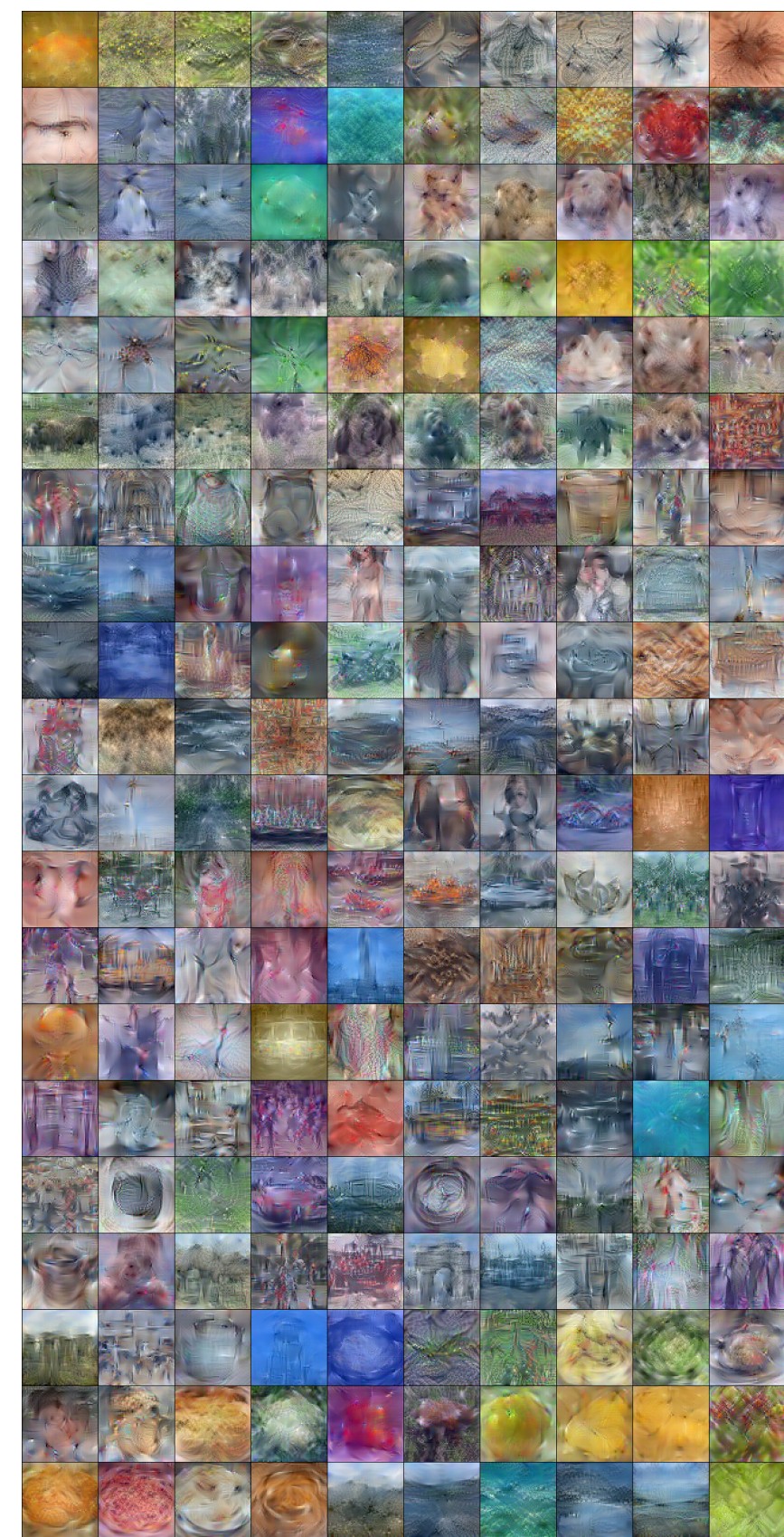

Figure 9: Distilled images of Tiny-ImageNet IPC1

