# OpenReview forum: "Prioritize Alignment in Dataset Distillation"
_ICLR.cc/2025/Conference — Submitted to ICLR 2025_

### Official Review · Reviewer_JQh4 · 2024-10-28

**Soundness:** 3
**Presentation:** 3
**Contribution:** 3
**Rating:** 6
**Confidence:** 4

**Summary:**

The paper introduces PAD, a method aimed at enhancing the compression of large datasets into compact synthetic datasets while preserving model performance. PAD addresses the challenge of misaligned information during distillation by aligning data through selective pruning of the target dataset and leveraging deep layers of agent models in the distillation process.

**Strengths:**

1. The identification of misalignment in dataset distillation and the proposal of PAD to address it is a valuable contribution.
2. The paper provides a clear motivation for the PAD method and supports its claims with extensive experiments.
3. The approach of prioritizing deep layer parameters for distillation is innovative and leads to improved performance.

**Weaknesses:**

1. It would be beneficial to prove the relationship between EL2N and the difficulty of training samples.
2. The paper could provide a more detailed comparison with other state-of-the-art methods, especially in terms of computational efficiency and scalability.
3. The theoretical analysis of why deep layer parameters are more suitable for distillation is lacking.
4. The paper does not discuss potential limitations or failure cases of the PAD method.

**Questions:**

1. How does PAD compare to other recent advances in dataset distillation in terms of computational resources and scalability?
2. Could the authors elaborate on the theoretical justification for prioritizing deep layer parameters?
3. Are there any scenarios where PAD might not perform as expected, and if so, how does the method handle such cases?

---

> ### Author Response · Authors · 2024-11-21
> **Response to Reviewer JQh4**
>
> Thanks for the comments. We provide explanations as follows.
>
> **W1: prove the relationship between EL2N and the difficulty**
>
> Thanks for the question. The definition of the EL2N score of a sample $(x, y) \in D$ is
> $$
> \mathbb{E}||p(w_t, x) - y||_{2}
> $$
> $p(w\_t, x) = \text{softmax}(f\_{t}(x))$ is the output of the model $f$, normalized by the softmax function at training time step $t$.  Clearly, this term can be viewed as the norm of the **error vector**. Thus, we reasonably assume that a large error vector implies a more difficult sample for the model to classify correctly.
>
> Also, our assumption is consistent with [6]
>
> **W2 & Q1: comparison with recent advances in terms of computational resources and scalability**
>
> Thanks for the question. PAD is proposed to alleviate the misalignment issue in previous matching-based methods. Thus, we mainly compare PAD with matching-based baselines in Table 1.
>
> **Scalability:**
>
> 1) **RDED**[7] is a non-optimization-based method enhanced strongly by **knowledge distillation** during evaluation. We already compared PAD with RDED in **Table 9**. We adopt the same evaluation strategy to make a fair comparison and show that PAD is superior, especially when the **IPC is larger**.
>
> 2. **SPEED**[8] is one representative of parameterization-based methods that prioritize proper parameterization of the synthetic dataset and reduce spatial redundancy. A comparison is shown below:
>
> **CIFAR-100:**
>
> | IPC   | 1        | 10       | 50       |
> | - | - | - | - |
> | SPEED | **40.4** | 45.9     | 49.1     |
> | PAD   | 28.8     | **48.4** | **56.2** |
>
> **Tiny ImageNet:**
>
> | IPC   | 1        | 10       | 50       |
> | - | - | - | - |
> | SPEED | **26.9** | 28.8     | 30.1     |
> | PAD   | 17.7     | **32.3** | **41.6** |
>
> As shown, SPEED outperforms PAD on IPC1 but **fails** to scale to large IPCs.
>
> Also, neither method reported performance on even larger IPCs. By contrast, PAD can achieve lossless performances on IPC500/1000. Therefore, **PAD's scalability on large IPCs is better.**
>
> **Resources:**
>
> 1. **RDED** is a non-optimization-based method, so it uses fewer computational resources than PAD.
> 2. **SPEED** is also built on a trajectory-matching backbone, so the computational cost is comparable.
>
> **W3 & Q2: theoretical analysis of why deep layer parameters are better**
>
> Thanks for the question. For trajectory-matching methods, the supervision of synthetic images is obtained by matching student and expert training trajectories. We claim the supervision produced by shallow-layer trajectories mainly introduces low-level, class-irrelevant information.
>
> This is based on previous works about the explanability of CNNs, such as Grad-CAM[9]. We provide examples of Grad-CAM of different layers in the link below.
>
> https://drive.google.com/file/d/1rR760U6dYtP4O4dIkYnAVesvHMVEcaeD/view?usp=sharing
>
> As observed, shallow-layer parameters mainly focus on class-irrelevant areas and can't learn semantical information.
>
> We then derive the relationship between the gradient of synthetic images and shallow-layer parameters.
>
> Denote $P$ as the set of pixels with semantical features and $Q$ as the rest. Denote the activation maps of shallow layers as $A^{\text{s}}$.  For $(i^p, j^p) \in P$ and $(i^q, j^q) \in Q$, based on the definition of Grad-CAM, we have:
>
> $$
> \alpha^{P} = \sum\_{i^p}\sum\_{j^p}  \frac{\partial y}{\partial A^{\text{s}}_{i^p, j^p}}
> $$
>
> $$
> \alpha^{Q} = \sum\_{i^q}\sum\_{j^q} \frac{\partial y}{\partial A^{\text{s}}_{i^q, j^q}}
> $$
> $\alpha$ is the gradient-based weight of the activation map, and $y$ is the model output.
>
> Since shallow layers focus on non-semantical areas, as introduced above, we have $\alpha^{P} \ll \alpha^{Q}$.
>
> Denote the trajectory-matching loss as $L$. When updating the synthetic data $D^{S}$ with shallow-layer parameters, we can compute the gradient as:
> $$
> \frac{\partial{L}}{\partial{D^{S}\_{i, j}}} = \frac{\partial{L}}{\partial{y}} \cdot \frac{\partial{y}}{\partial A^{\text{s}}\_{i, j}} \cdot \frac{\partial A^{\text{s}}\_{i, j}}{\partial D^{S}\_{i, j}}
> $$
> Consequently, if $(i, j) \in P$, the term $\alpha^{P}\_{i, j} =\frac{\partial{y}}{\partial A^{\text{s}}\_{i, j}}$ will be small; when $(i, j) \in Q$, $\alpha^{Q}\_{i, j} = \frac{\partial{y}}{\partial A^{\text{s}}\_{i, j}}$ is much larger. This shows that supervision produced by shallow-layer parameters typically **downplays semantical areas** of the synthetic images while **emphasizes class-irrelevant ones**.
>
> By contrast, the relationship between gradient and class-relevant information is positive on deep layers.
>
> Therefore, we propose to drop more supervision produced by shallow layers as the IPC increases.
>
> **W4 & Q3: potential limitations or failure**
>
> Thanks for the question. PAD is proposed to improve matching-based methods for dataset distillation. Therefore, PAD is not directly applicable to methods that don't follow the two-step workflow, information extraction and information embedding, such as kernel-based methods.

---

> ### Author Response · Authors · 2024-11-24
> **To save reviewer's time, we put a summary of rebuttal**
>
> Dear Reviewer JQh4,
>
> Thanks so much again for the time and effort in our work. Considering the limited time available and to save the reviewer's time, we summarize our responses here.
>
> 1. **[Prove the relationship between EL2N and the difficulty of training samples]**
>
>    **Response**: EL2N can be viewed as "error vector". This relationship is also used in previous works.
> 2. **[Comparison with recent advances in terms of computational resources and scalability]**
>
>    **Response**:
>    - We already compared with RDED in the submission. In the rebuttal, we compare with another recent SOTA, SPEED.
>    - We show that PAD can scale to larger IPCs more effectively.
> 3. **[Theoretical analysis of why deep layer parameters are more suitable for distillation]**
>
>    **Response**:
>    - Our theoretical support comes from previous works about the explainability of convolutional neural networks, such as Grad-CAM.
>    - We show the derivation of the effect of updating synthetic images using shallow-layer trajectories based on Grad-CAM in the rebuttal.
> 4. **[Potential limitations or failure cases of the PAD method]**
>
>    **Response**:
>    - For methods that don't follow the two-step workflow, information extraction and information embedding, PAD is not directly applicable. For example, kernel-based methods.
>
> Since the discussion stage is closing soon, may I know if our rebuttal addresses the concerns? If there are further concerns or questions, we are more than happy to address them. Thanks again for taking the time to review our work and provide insightful comments.
>
> Best Regards,
>
> Authors

---

> > ### Comment · Reviewer_JQh4 · 2024-11-24
> > **Raising Score Following Rebuttal**
> >
> > I appreciate the authors' detailed clarifications. They have thoroughly addressed most of my concerns, and based on their additional explanations and efforts during the rebuttal, I have decided to raise my score from 5 to 6.

---

> > > ### Author Response · Authors · 2024-11-24
> > > **We appreciate the support**
> > >
> > > Thanks very much for the support. We are happy to see that we have addressed your concerns. Your valuable feedback is very helpful for us in improving our work.
> > >
> > > If you have any further questions, we are more than happy to provide a detailed explanation. Thanks again for supporting our work.

---

### Official Review · Reviewer_3Mn9 · 2024-10-31

**Soundness:** 3
**Presentation:** 3
**Contribution:** 2
**Rating:** 5
**Confidence:** 4

**Summary:**

This paper explores the role of alignment in both the information extraction and embedding stages within the dataset distillation process.
During the information extraction stage, alignment is achieved by selecting subsets of the dataset based on difficulty. For settings with a low images-per-class (ipc) count, incorporating a higher proportion of ‘easy’ data proves effective. Conversely, in high ipc settings, utilizing a larger portion of ‘hard’ data enhances distillation effectiveness. The EL2N score, analogous to model confidence, is employed as a difficulty metric.
In the information embedding stage, alignment is achieved by prioritizing deeper over shallower layers. Deeper layers are more adept at learning semantic information, resulting in a higher-level representation within synthetic data and more efficient distillation.

**Strengths:**

1. One advantage of this method is that it can be applied on top of other distillation methods, such as MTT and DM. As seen in cross-architecture analyses, it’s a general approach that enhances performance in a scalable, model- and dataset-agnostic way. Experimental results show that accuracy improved over traditional methods (Table 1), with increases across multiple datasets and architectures (Table 2).

2. It seems the straightforward method is supported by detailed analysis and experiments. Experiments (Tables 4, 5 / Figure 5) effectively support the hypothesis that removing easy examples and using deeper layers improves performance, making the qualitative results intuitively understandable (Figures 4, 6).

3. While the writing isn’t exceptionally well-crafted, it is organized in a readable and accessible manner.

**Weaknesses:**

1. I believe the contribution of this method is limited. The proposed method is very simple, and while it is supported by empirical experimental results, it lacks theoretical justification. Additionally, as mentioned in the paper, the data selection and distillation approach was already introduced by BLiP (Xu et al., 2023). Although the paper presents experimental results showing superior performance to this approach (Table 5.b), the metric for selecting difficulty (EL2N) was also previously proposed.

2. The method seems somewhat sensitive to hyperparameters (AEE, IR). Accuracy varies depending on these hyperparameters (Table 4.a), with differences comparable to the performance increase over other methods in the main table. Additionally, as seen in Table 10, optimal hyperparameters change dynamically across datasets and ipc values, raising questions about the method’s stability. If it is indeed sensitive, hyperparameters may need to be tuned for each architecture and dataset, which raises concerns about the method’s practicality.

3. Although the paper claims that performance improves when used alongside various distillation techniques, as mentioned in the limitations section, due to computing resource constraints, experiments were conducted only with DM and DC (and DATM in the main table).

Overall, the paper demonstrates that a simple method can enhance distillation performance and can be used in a model- and architecture-agnostic manner. The experiments and logical development are well-executed; however, I believe the contribution of the method itself is limited, and it appears to be sensitive to hyperparameters. Therefore, I would rate it as [5: marginally below the acceptance threshold].

**Questions:**

1. In the main table (Table 1), it appears that multiple experiments were conducted to obtain an average, but subsequent tables present experimental values as single data points. I couldn't find any information in the paper regarding whether the results for each experiment were averaged, and I’m curious about this.

2. The study exclusively uses deep layers; are shallow layers entirely unnecessary? Shallow layers likely play a role in producing a good embedding space, so is there a way to leverage them effectively?

---

> ### Author Response · Authors · 2024-11-21
> **Response to Reviewer 3Mn9**
>
> Thanks for the valuable comments. We make clarifications as follows.
>
> **W1: I believe the contribution of this method is limited.**
>
> Thanks for the concern. We argue that although our method seems simple, it comprehensively analyzes the limitations of existing matching-based methods and proposes an easy but effective approach. More importantly, our solution is generalizable to many existing approaches.
>
> For BLiP[5], not only is it less effective than PAD, but it is also less efficient and less generalizable. This is because it applies static data pruning. Thus, expert trajectories need to be trained on a case-by-case basis due to different pruning ratios. PAD allows the same set of trajectories to be used for **all IPCs** in one dataset. It reduces the overhead of training trajectories multiple times.
>
> | Method | IPC 1                | IPC 10               | IPC 50               |
> | ------ | -------------------- | -------------------- | -------------------- |
> | BLiP   | Retrained trajectory | Retrained trajectory | Retrained trajectory |
> | PAD    | Shared trajectory    | Shared trajectory    | Shared trajectory    |
>
> **W2: The method seems somewhat sensitive to hyperparameters (AEE, IR).**
>
> Thanks for raising this concern. We would like to make the following clarifications:
>
> - These two hyper-parameters are easy to tune since they only affect small IPCs.
> - They don't need to be changed according to the target datasets. Through experiments, we find setting IR=75% and AEE=40 generalize well across various datasets. In all experiments reported in paper, we use these settings by default except only change AEE to 20 on CIFAR-10.
> - Generally, 75%\~80% for IR and 20\~40 for AEE are good settings as the performance doesn't change too much within these ranges. We show ablation results on the other two benchmarks in the tables below:
>
> **CIFAR-100, IPC10**
>
> | IR   | AEE=20 | AEE=30 | AEE=40   |
> | :--- | :----- | :----- | :------- |
> | 50%  | 47.8   | 47.7   | 47.7     |
> | 75%  | 48.3   | 48.1   | **48.4** |
> | 80%  | 48.2   | 48.3   | 48.3     |
>
> **Tiny ImageNet, IPC1**
>
> | IR   | AEE=20 | AEE=30 | AEE=40   |
> | :--- | :----- | :----- | :------- |
> | 50%  | 16.9   | 17.2   | 17.4     |
> | 75%  | 17.3   | 17.1   | **17.7** |
> | 80%  | 17.2   | 17.5   | 17.6     |
>
> Thanks again for raising this concern. We have added the above results and analysis to the Appendix.
>
> **W3: due to computing resource constraints, experiments were conducted only with DM and DC**
>
> Thanks for the concern. DC[1], DM[2], and MTT[3] are three foundational matching-based methods of DD. We think the effectiveness of our ablation on these methods is sufficient to demonstrate the limitations of existing matching-based methods. In Section 5.4, we show that PAD is also effective in more recent methods, such as SRe2L[4].
>
> **Q1: subsequent tables present experimental values as single data points.**
>
> We are sorry for any confusion. In the subsequent ablation experiments, we only report the **mean** values of multiple runs. This is because we want to show the efficacy of our two modules or performances of different hyper-parameters, so the standard deviation is not very relevant. We have included a clarification in the revision to avoid confusion. Thanks for pointing out.
>
> **Q2: The study exclusively uses deep layers; are shallow layers entirely unnecessary**
>
> Thanks for the question.
>
> For the first part of the question, we want to clarify that we are **not** exclusively using deep layers. Instead, we dynamically drop different ratios of shallow-layer parameters according to different IPCs. An example on CIFAR-10 is shown in the table below (we regard the first half of model parameters as shallow layers, and the second half as the deep layers).
>
> | IPC1                                             | IPC50                                               | IPC500                                      |
> | ------------------------------------------------ | --------------------------------------------------- | ------------------------------------------- |
> | **Preserve all** shallow-layer parameters (100%) | **Preserve part** of shallow-layer parameters (50%) | **Drop all** shallow-layer parameters. (0%) |
>
> For the second part, shallow layers are **not** entirely unnecessary. We filter out shallow-layer parameters because they typically learn easy and common patterns. Although these patterns introduce misaligned information to large compressing ratios, they are useful for small compressing ratios. This is because the capacity of a small compressing ratio is much smaller, so low-level patterns are sufficient. Thus, PAD adjusts the dropping ratio according to different IPCs.
>
> We are happy to answer any of further questions you may have.

---

> ### Author Response · Authors · 2024-11-24
> **To save reviewer's time, we put a summary of rebuttal**
>
> Dear Reviewer 3Mn9,
>
> Thanks so much again for the time and effort in our work. Considering the limited time available and to save the reviewer's time, we summarize our responses here.
>
> 1. **[Limitated contribution]**
>
>    **Response**: PAD comprehensively analyzes the limitations of existing matching-based methods and proposes an easy but effective approach. It is not an incremental work.
> 2. **[Sensitive to hyper-parameters]**
>
>    **Response**:
>    - PAD introduces two hyper-parameters during information extraction and they are easy to tune.
>    - We provide additional experimental results to show that 75%~80% for IR and 20~40 for AEE are good settings as the performance doesn't change too much within these ranges.
> 3. **[More experiments except DC and DM]**
>
>    **Response**:
>    - DC, DM, and MTT are representative enough since existing matching-based methods are mostly built on these three foundational methods.
>    - In Section 5.4, we show that PAD is also effective in more recent methods, such as SRe2L.
> 4. **[Subsequent tables present experimental values as single data points]**
>
>    **Response**:
>    - We have added clarification and updated the revision. Thanks for pointing it out.
> 5. **[The study exclusively uses deep layers]**
>
>    **Response**:
>    - PAD does not exclusively use deep layers. Instead, it dynamically drops different ratios of shallow-layer parameters according to different IPCs.
>    - Shallow layers are not entirely unnecessary. On small IPCs, we find low-level patterns embedded by shallow-layer parameters are sufficient.
>
> Since the discussion stage is closing soon, may I know if our rebuttal addresses the concerns? If there are further concerns or questions, we are more than happy to address them. Thanks again for taking the time to review our work and provide insightful comments.
>
> Best Regards,
>
> Authors

---

> ### Author Response · Authors · 2024-11-24
> **Your feedback is valuable to us**
>
> Dear Reviewer,
>
> Thanks again for providing the constructive review. We tried our best to address your concerns point by point so that you can have a better understanding of our work.
>
> However, since the discussion period is closing soon, we are eager to hear from you. Your further feedback is valuable for us to continue improving our work. If you still have other questions, we are more than willing to explain in detail.
>
> Looking forward to your reply.

---

> ### Author Response · Authors · 2024-11-25
> **Looking forward to your reply**
>
> Dear Reviewer,
>
> Since the discussion period is going to end in one day, we really want to hear from you and see if we have addressed your concerns. Please participate in the discussion because your feedback is important for us to improve our work.
>
> Thank you.

---

### Official Review · Reviewer_BA9i · 2024-11-03

**Soundness:** 3
**Presentation:** 3
**Contribution:** 2
**Rating:** 6
**Confidence:** 5

**Summary:**

The paper introduces "Prioritize Alignment in Dataset Distillation" (PAD), a method improving dataset distillation by focusing on two main strategies. First, it adjusts data selection based on different Information Compression Ratios (ICRs) to match the required difficulty levels. Second, it enhances results by distilling only from deep-layer network parameters. The effectiveness of PAD is validated through experiments on standard datasets like CIFAR-10, CIFAR-100, and Tiny ImageNet, showing improvements over current DD methods.

**Strengths:**

1. The paper demonstrates substantial improvements in DD through its experimental results in TAB. 1, showing that the PAD method significantly enhances the effectiveness of dataset distillation.
2. PAD proves to be highly adaptable and generalizes well across multiple datasets, indicating that the method can be effectively applied to various dataset distillation tasks with consistent success.

**Weaknesses:**

1. There are some formatting issues in the manuscript that need attention. In **Table 5**, the annotations (a) and (b) appear to be reversed, which could confuse readers. Additionally, a period is missing at the end of line 485 after "smaller IPCs." Please correct these to enhance the clarity and professionalism of the document.
2. There seems to be a mismatch between the issues presented in Section 2.1 and the methods proposed in Section 3.2. The former discusses a static selection method, while the latter introduces a dynamic approach. Could the authors clarify the connection between these sections to ensure the consistency of the methodology described?

**Questions:**

1. The manuscript employs DATM as a baseline, which, as I understand it, requires the pre-training of numerous agent models on the original dataset to record training trajectories. As discussed in w2, given the dynamic dataset scheduler nature of the process described in your methodology, does this imply the need to train additional agent models? It would be beneficial for readers if the authors could provide more detailed insights into the implementation specifics of this approach.

---

> ### Author Response · Authors · 2024-11-21
> **Response to Review BA9i**
>
> Thanks for the valuable comments. We make responses as follows:
>
> **W1: There are some formatting issues in the manuscript that need attention.**
>
> We are very sorry for these two careless mistakes and have updated the revision. For the camera-ready version, we will double check the manuscipt and make sure all such mistakes (if any) are corrected. Thanks for pointing out.
>
> **W2: There seems to be a mismatch between the issues presented in Section 2.1 and the methods proposed in Section 3.2.**
>
> Thanks for raising this concern. We break the weakness into two parts.
>
> **W2.1:** In Section 2.1,the method is **static**.
>
> **A:** In this section, we conducted ablation experiments to show two observations:
>
> 1. On small IPCs, removing various ratios of hard samples before distillation improves the performance.
> 2. On large IPCs, removing various ratios (within a reasonable range) of easy samples before distillation improves the performance.
>
> Both observations lead to our conclusion that misalignment exists in the information extraction stage of previous matching-based DD methods.
>
> **W2.2:** In Section 3.2, the method is **not stati**c, mismatching Section 2.1.
>
> **A:**  Our method is dynamic because of efficiency consideration. PAD is built on trajectory-matching methods which need to train expert trajectories. Based on our empirical analysis in Section 2.1, information of hard/easy samples should be removed from trajectories for small/large IPCs. However, it is **not efficient** to train multiple sets of trajectories with different sets of data.
>
> What PAD does is to **extract information of different difficulty levels at different epochs in only one trajectory**, which is achieved by our scheduler in Section 3.2. This is correct because we only match a particular range of epochs for each IPC. By doing so, we can filter out misaligned information in the extraction stage **for all IPCs** by training **only one set** of expert trajectories.
>
> **Q1: Does this imply the need to train additional agent models**
>
> **A:** Thanks for the question. Please correct us if we are wrong. The concern you have is whether we need to train multiple sets of trajectories according to our method. The answer is **no** because our scheduler is designed to allow us to train **only one set** of expert trajectories but filter out misaligned information for all IPCs at the same time. The explanation is similar to the above: each IPC matches a specific range of epochs, so the scheduler can dynamically filter out difficulty-misaligned data at different stages. In the table below, we show an example of CIFAR-10 to illustrate how one set of trajectories is shared among different IPCs.
>
> We are happy to continue the discussion if we misinterpret your question.
>
> | IPC1                | IPC10               | IPC50               | IPC500              | IPC1000             |
> | ------------------- | ------------------- | ------------------- | ------------------- | ------------------- |
> | Shared trajectories | Shared trajectories | Shared trajectories | Shared trajectories | Shared trajectories |

---

> > ### Comment · Reviewer_BA9i · 2024-11-25
> >
> > Thank you to the authors for the clarification and detailed explanation. The insights provided are valuable and well-reasoned. I will maintain my original score.

---

> > > ### Author Response · Authors · 2024-11-26
> > > **We appreciate the support**
> > >
> > > Dear Reviewer,
> > >
> > > Thanks for replying. We appreciate the support. If you still have any other questions, please don't hesitate to ask us. We will explain every detail.
> > >
> > > Best,
> > >
> > > Authors

---

> ### Author Response · Authors · 2024-11-24
> **To save reviewer's time, we put a summary of rebuttal**
>
> Dear Reviewer BA9i,
>
> Thanks so much again for the time and effort in our work. Considering the limited time available and to save the reviewer's time, we summarize our responses here.
>
> 1. **[Formatting issues]**
>
>    **Response**: We appreciate the reviewer for identifying these mistakes. We have corrected them and updated the revision.
> 2. **[Mismatch between Section 2.1 and Section 3.2]**
>
>    **Response**:
>    - Section 2.1 presents an ablation study to empirically demonstrate the exitance of misaligned information in both information extraction and information embedding stages.
>    - Section 3.2 illustrates how we implement PAD on a trajectory-matching backbone. Our dynamic data scheduler is designed to avoid trajectory re-training.
> 3. **[Do we need to train additional agent models]**
>
>    **Response**: We don't need to train multiple sets of expert trajectories. Our FIEX module can filter out misaligned information in only one set of trajectories.
>
> Since the discussion stage is closing soon, may I know if our rebuttal addresses the concerns? If there are further concerns or questions, we are more than happy to address them. Thanks again for taking the time to review our work and provide insightful comments.
>
> Best Regards,
>
> Authors

---

> ### Author Response · Authors · 2024-11-24
> **Your feedback is valuable to us**
>
> Dear Reviewer,
>
> Thanks again for providing the constructive review. We tried our best to address your concerns point by point so that you can have a better understanding of our work.
>
> However, since the discussion period is about to end, we are eager to hear from you. Your further feedback is valuable for us to continue improving our work. If you still have other questions, we are more than willing to explain in detail.
>
> Looking forward to your reply.

---

### Official Review · Reviewer_QoJt · 2024-11-06

**Soundness:** 3
**Presentation:** 3
**Contribution:** 2
**Rating:** 3
**Confidence:** 4

**Summary:**

This paper focuses on the task of dataset distillation, which aims to compress a large dataset into a much more compact synthetic dataset while maintaining the performance of trained models. Existing methods rely on an agent model to extract and embed information from the target dataset into the distilled version. However, the authors identify that current approaches often introduce misaligned information during the extraction and embedding stages, which degrades the quality of the distilled dataset.

To address this, the authors propose Prioritize Alignment in Dataset Distillation (PAD), a method that aligns information from two main perspectives: Dataset Pruning and Deep Layer Utilization. This simple yet effective strategy helps filter out misaligned information, resulting in significant improvements for mainstream matching-based distillation algorithms. Additionally, when applied to trajectory matching, PAD achieves state-of-the-art performance across various benchmarks, showcasing its effectiveness.

**Strengths:**

1. Figures 2 and 3 provide valuable insights into the effects of removing both simple and difficult samples, as well as the impact of shallow parameters on model performance.

2. The authors present a comprehensive experimental analysis on a small-scale dataset, including ablation studies, hyperparameter analysis, and related discussions.

**Weaknesses:**

1. The author's approach appears to be more incremental, incorporating only minor enhancements to DATM. Firstly, Section 3.1 serves as a review of previous work, and Equation (4) in Section 3.3 shows only slight modifications from Equation (3) in DATM. Additionally, Section 3.2 seems to function more as a heuristic for selecting difficult samples. Overall, the method introduces only two additional techniques compared to DATM, lacking a significant breakthrough in terms of innovation.

2. in Table 2, most of the average improvements over the comparable method, DATM, are less than 1 point, indicating that the performance still requires further enhancement.

**Questions:**

The thesis would benefit from a more structured presentation, where the authors are encouraged to list observations and analyses in Chapter 2 in a systematic manner, similar to DTAM. Additionally, the content should be included in the methods section as a motivation or exploration segment to strengthen the logical flow and context of the proposed

---

> ### Author Response · Authors · 2024-11-21
> **Response to Reviewer QoJt**
>
> Thanks for the comments. However, we want to correct the term "Data Pruning and Deep Layer Utilization" in the summary as it **does not accurately describe our work.**
>
> **Data:** What PAD discovers from the previous matching-based methods is that during information extraction, different samples should be used for IPCs with different capacities. This is **different from simply pruning the dataset**, which only focuses on data redundancy but fails to customize it for different compressing ratios. Also, **re-training** expert trajectories is required in their methods. PAD efficiently fits the expected information difficulty levels for different IPCs with **only one set** of trajectories.
>
> **Supervision:** What PAD does is **not utilizing deep layers only**. Based on previous works on the explanability of CNNs, we discover that the supervision produced by matching shallow-layer parameters injects low-level and class-irrelevant information into synthetic data. As the IPC becomes larger, such information becomes more harmful. Therefore,  we propose the FIEM module that **dynamically drops shallow-layer parameters** during matching according to different IPCs.
>
> **W1 & Q1: The author's approach appears more incremental. The thesis would benefit from a more structured presentation**
>
> Thanks for the concern. **We first elaborate on the structure of this work again:**
>
> 1. We summarize the general workflow of existing matching-base methods into two steps: information extraction and information embedding. Foundational methods, including DC[1], DM[2], and MTT[3], all follow this workflow. (Section 1)
> 2. We identify misaligned information in both steps. Misalignments come from the ignorance of the different capacities of different IPCs. (Section 1)
>    a) For information extraction, small IPCs prefer simpler patterns, while large IPCs prefer more semantical ones.  Thus, different IPCs should extract information from different subsets of the dataset.
>    b) For information embedding, more supervisions produced by shallow-layer parameters of the surrogate model should be dropped as the IPC increases since they usually introduce low-level information.
> 3. Our observations are supported by ablation studies on three matching-based baselines.
>    a) For information extraction, within a reasonable range, removing hard samples helps improve the distillation performance, and removing easy samples improves the distillation performance in large IPC settings. (Section 2.1)
>    b) For information embedding, removing supervision produced by shallow-layer parameters improves performance, especially for large IPCs.  (Section 2.2)
> 4. With the empirical analysis, we propose to prioritize alignment for matching-based methods to build PAD on a trajectory-matching backbone. (Section 3)
>
> We believe the logic of our paper is clear and concise. The limitation we discover is universally present in matching-based methods. Our proposed method is easy, effective, and generalizable.
>
> Now, we break down the question and answer point by point.
>
> **W1.1: Equation (4) in Section 3.3 shows only slight modifications from Equation (3) in DATM.**
>
> **A:** Our solution to drop misaligned parameters is straightforward, effective and can be easily applied to other matching-based methods. We argue that complex modification doesn't always lead to improvement, and it may reduce the generalizability.
>
> **W1.2: Section 3.2 seems to function more as a heuristic for selecting difficult samples**
>
> **A:** Again, we want to correct the reviewer's description. Section 3.2 is **not** just about selecting difficult samples like what pruning does. Instead, we use easier samples to train early trajectories and harder samples to train late trajectories. Also, the scheduler allows us to train only one set of trajectories instead of re-training whenever the subset of data is changed.
>
> **Q1: The thesis would benefit from a more structured presentation like DATM**
>
> **A:** We want to argue that the structure of DATM is **not good for our paper**. In Section 2, what we discover are **general** limitations of matching-based methods. Therefore, we use a separate section to provide details. DATM targeted only on trajectory-matching so it describes the motivation in the method section.
>
> We appreciate the reviewer's advice on enhancing the analysis in Section 2. We will make improvements in the camera-ready version.
>
> **W2: In Table 2, most of the average improvements over DATM are less than 1**
>
> The results in Table 2 are cross-architecture evaluation results. These results are to show that our distilled datasets can also bring comparable performances on other models. **However, the main focus of this paper is not cross-architecture generalization**. We show that PAD is better than DATM in many cases and comparable in other cases. We believe these results can already demonstrate the generalizability of PAD on unseen models.  **For the distillation performance, please refer to Table 1**.

---

> > ### Comment · Reviewer_QoJt · 2024-11-22
> > **Further Discussion**
> >
> > First of all, I appreciate the authors' response. Here is my further discussion:
> >
> > The performance improvement is marginal. Not only in Table 2 but also in Table 1, the performance difference between DATM and PAD is only slightly above 1, which I do not consider a significant improvement.
> >
> > In my opinion, the contributions are incremental, both in terms of performance and methodology. Unfortunately, my perspective on this matter remains unchanged.

---

> > > ### Author Response · Authors · 2024-11-22
> > > **Response to Review QoJt**
> > >
> > > Thanks for the reply. However, we **can't** agree with you.
> > >
> > > For the performance: Since our baseline, DATM has achieved **lossless results**  in some cases, we believe an average improvement above 1% already demonstrates our contribution. In the previous review, you think improvements below 1% in Table 2 are unacceptable. But after we informed you that Table 1 is our main performance, you changed your opinion and stated that improvements above 1% are also unacceptable to you. We believe that this contradiction reflects the **unprofessionalism of the review opinions**.
> > >
> > > For the methodology: We have **repeatedly** emphasized that PAD is proposed to solve **a general limitation that exists in matching-based methods**. We build PAD on DATM, but PAD is **not just incremental**. We have also shown PAD's efficacy in the recent advance. We argue that the term "incremental" comes from the reviewer's poor understanding of our work.

---

> ### Author Response · Authors · 2024-11-25
> **Looking forward to further discussion**
>
> Dear Reviewer,
>
> We understand that you still have concerns about our work. We are eager to hear **details**. Since the discussion period is less than one day, we encourage you to tell us any questions you still have. We promise to address them in detail. Thank you.
>
> Best Regards

---

### Author Response · Authors · 2024-11-21
**General Response to All Reviewers**

We are grateful for the hard work done by all reviewers. Your comments are important for us to make further improvements.

We have updated the revision and marked modifications in red.

We try our best to answer all questions point by point in great detail. If you still have questions, don't hesitate to continue the discussion with us. We are more than willing to clear up any doubts you may have.

Below are our references used in responses:
[1] *Dataset Condensation with Gradient Matching*, ICLR 2020

[2] *Dataset Condensation with Distribution Matching*, CVPR 2021

[3] *Dataset Distillation by Matching Training Trajectories*, CVPR 2023

[4] *Squeeze, Recover and Relabel: Dataset Condensation at ImageNet Scale From A New Perspective*, NeuIPS 2023

[5] *Distill Gold from Massive Ores: Efficient Dataset Distillation via Critical Samples Selection*, ECCV 2024

[6] *Beyond neural scaling laws: beating power law scaling via data pruning*, NeuIPS 2022

[7] *On the Diversity and Realism of Distilled Dataset: An Efficient Dataset Distillation Paradigm*, CVPR 2024

[8] *Sparse Parameterization for Epitomic Dataset Distillation*, NeuIPS 2023

[9] *Grad-CAM: Visual Explanations from Deep Networks via Gradient-based Localization*, ICCV 2017

---

### Meta-Review · Area_Chair_fgXv · 2024-12-17

**Metareview:**

This submission received two negative sores and two positive scores after rebuttal. After carefully reading the paper, the review comments, the AC can not recommend the acceptance of this submission, as the average score is under the threshold bar and the concern about the technical novelty remains. The AC also recognizes the contributions confirmed by the reviewers, and encourages the authors to update the paper according to the discussion and submit it to the upcoming conference.

**Additional Comments On Reviewer Discussion:**

After discussion, while reviewer # BA9i and #JQh4 thought that the responses fully addressed their concerns reviewer#oxLB and #3Mn9 did not responded to the rebuttal and kept the original negative scores.After discussion, while reviewer # BA9i and #JQh4 thought that the responses fully addressed their concerns reviewer#oxLB and #3Mn9 did not responded to the rebuttal and kept the original negative scores.

---

### Decision · Program_Chairs · 2025-01-22

Reject